# Experiences of interventions and rehabilitation activities in connection with return-to-work from a gender perspective. A focus group study among employees on sick leave for common mental disorders

Lotta Nybergh[1]*, Gunnar Bergström[1,2‡], Irene Jensen[1‡], Therese Hellman[3]

1 Unit of Intervention and Implementation Research for Worker Health, Institute of Environmental Medicine, Karolinska Institute, Stockholm, Sweden, 2 Department of Occupational Health Sciences and Psychology, Centre for Musculoskeletal Research, University of Gävle, Gävle, Sweden, 3 Unit of Occupational and Environmental Medicine, Department of Medical Sciences, Uppsala University, Uppsala, Sweden

☺ These authors contributed equally to this work.
‡ These authors also contributed equally to this work.
* lotta.nybergh@ki.se

**Data Availability Statement:** Since the data is based on transcribed interviews, the data cannot be completely anonymized and can therefore not be

## Abstract

### Background

Common mental disorders present the main reason for registered sick leave in Sweden today, and women are at a higher risk of such sick leave than men. The aim of this paper is to explore how employees on sick leave for common mental disorders experience interventions and rehabilitation activities during return-to-work, as well as to explore similarities and differences between the experiences of the interviewed women and men.

### Material and methods

A qualitative design was applied with semi-structured focus group interviews. Seven focus groups were conducted with a total of 28 participants (13 women and 15 men). The focus group discussions were audiotaped and transcribed verbatim, and data analyzed with conventional content analysis. Similarities and differences in the women's and men's experiences were written down in reflective notes during all steps of the analysis.

### Results

The results comprise of one main category, "To be met with respect and recognition", and subcategories at two levels. Both similarities and differences emerged in how women and men sick-listed because of common mental disorders experienced return-to-work interventions and rehabilitation activities. It was important for both women and men to be met with respect and recognition, which was essential to all forms of help that the participants discussed during the focus group interviews. Women expressed a need for home-related interventions, whereas men expressed a need for organizational interventions to counter

shared openly; although names, places and the like have been removed from the transcripts, the participants speak of situations and sensitive experiences within specific contexts and with a level of detail that would pose a risk for the participants to be recognized. The participants have also not given their consent for the interview transcripts to be freely shared. Requests by certified researchers for access to the data should therefore be made to the Research and Data Office at the Karolinska Institute, via rdo@ki.se. If permitted by law and ethical approval, which is decided on a case by case basis, the data can be shared.

**Funding:** This study was funded by the Swedish Research Council for Health, Working Life and Welfare (GB, grant number 2014-0742), and by the Swedish Social Insurance Agency (GB, LN, grant number 059169 – 2015). https://forte.se/en/ https://www.forsakringskassan.se/sprak/engelska The funders had no role in study design, data collection and analysis, decision to publish, or preparation of the manuscript.

**Competing interests:** The authors have declared that no competing interests exist.

feelings of resignation at work. Women could also more easily understand their mental health condition as compared with men.

## Conclusion

A key implication of this study is that research on interventions and rehabilitation activities during return-to-work among employees on sick leave for common mental disorders should consider whether the findings are relevant equally to both women and men. Similarly, return-to-work professionals may need to consider possible differences among women and men on sick leave for common mental disorders, and to further customize offered interventions and rehabilitation activities. Doing so may help enhance the effectiveness of such interventions.

## Background

Sickness-related absence for common mental disorders (CMD), such as depression, anxiety and stress-related disorders, affects workers' health, economic security, professional outlook, identity and sense of self-worth [1, 2]. It also compromises job performance and increases the risk for early withdrawal from the job market [3, 4]. In addition to the significant consequences of individual suffering, CMD entail substantial economic costs for the employer and for the society in terms of lost productivity and compensation-related costs [5, 6].

CMD-related sickness absence is more prevalent among women than men: in Sweden, the prevalence in 2018 was 53% for women and 40% for men [7]. This difference has been attributed to the so-called horizontal as well as vertical gender segregation. The former refers to the understanding that female-dominated professions tend to entail factors known to increase mental ill-health [8], whereas the latter refers to the fact that men occupy leadership positions more often and have higher opportunities to influence their working situations [9]. In addition, home-related factors have been investigated. For example, studies have found that women's risk for sickness absence increases after the arrival of the first child [10, 11]. Finally, some studies have attributed the disparity in sickness absence to gender norms. For example, a study conducted among women with mental health problems reported that women found it difficult to draw boundaries in an unrelenting work environment, which stemmed from a perception of having a lower social status in relation to a male-dominated environment or a male manager [12]. Further, a recent report published in Sweden found that gender norms could explain why women with mental health problems were 30% more likely to be on sickness absence than men even when their work ability was estimated to be at the same level [13]. The authors argue that women may exhibit more preventive behaviors than men and are more receptive toward sickness absence in encounters with the doctors than men, with the latter preferring more medical treatments. Health professionals may also not be able to recognize mental ill health as well in men as in women [14].

Given the high prevalence and various consequences of CMD-related sickness absence for the worker, employer and society, interventions and rehabilitation activities that aim to expedite return-to-work (RTW) have received growing attention. Recent reviews on mainly work-based interventions for employees with CMD have concluded that they appear to improve such aspects as the duration of the absence, work functioning, quality of life and economic outcomes [6, 15–19]. However, the reviews also reveal some limitations in the effectiveness of the

interventions (e.g. in returning to work with full hours), which calls for further investigation of workers' perceptions and experiences of interventions to RTW. Notably, despite the sex-discrepancy in CMD-related sickness absence, studies on RTW–including the reviews referred to above [6, 15–19]–rarely apply a gender perspective [20]. One of the reviews did highlight the need for future studies that include more female samples to help identify possible gender-specific effects of RTW interventions [18].

Although the literature is scarce, some studies do report gender differences in rehabilitation activities for employees on sickness absence because of CMD. For example, one study found that women on sick leave for CMD were less likely than men to participate in a sickness-absence interview with their employer or to receive care from a psychiatrist [21]. The women were also less likely than men to feel that officers took their condition seriously during RTW, although the contrary was found in a study among women and men with musculoskeletal disorders [22]. A gender-focused review on sickness absence highlighted the need for additional research into gender-relevant aspects in diagnostics and treatment [23].

Most studies on RTW interventions and rehabilitation activities, as well as their effects among employees with CMD, apply a quantitative study design. However, qualitative studies are particularly well-suited to illuminating the experiences of those involved in the RTW process to deepen our understanding of the complexities involved and to help interpret the results of quantitative studies [24]. For example, a longitudinal qualitative study found that an individual approach of "being seen" and "feeling met" by the RTW professionals determined the extent to which the intervention and other RTW activities were considered to be helpful [25]. Although the study included women and men, it did not compare their experiences to see whether they might differ. This is also the case with a meta-synthesis of qualitative studies on RTW among employees with CMD, which identified obstacles and opportunities in relation to the handling of individual demands, social support, accommodations at the workplace and contact with the systems (e.g. social insurance offices) [24]. A study that focused on employees sick-listed because of musculoskeletal disorders did compare women's and men's experiences of how the domestic arena affected RTW and found that domestic demands impeded women's RTW. Although the authors did not directly explore the participants' experiences of RTW activities, they argued that domestic strain needs to be considered in the rehabilitation processes of women [26]. Studies that compare women's and men's experiences of interventions, rehabilitation activities and help offered during RTW among employees on sick leave for CMD could help guide the development of interventions in a way that is sensitive to the needs of both genders.

The aim of this paper is thus to explore how employees on sick leave for CMD experience interventions and rehabilitation activities during RTW, as well as to explore similarities and differences between the experiences of the interviewed women and men.

## Material and methods

The methods section follows the consolidated criteria for reporting qualitative research [27]. This section is similar to that reported in another manuscript with a distinct but intersecting aim [28], as they belong to the same core project (see below). Furthermore, an abbreviated version of the results of this study have been published by the project's funder in a report written in Swedish [29].

### Design

A qualitative design with purposive data sampling was applied with semi-structured focus group interviews [30]. The interviews and interview guide were designed to answer two

distinct but intersecting aims, of which the first (on the experiences of demands and resources during RTW) is presented in a separate article [28] and the other (on the experiences of interventions and rehabilitation activities) in the current paper.

## Data collection and participants

This qualitative interview study is part of a larger project that aims to evaluate an intervention by the occupational health services (OHS) directed at employees with CMD or stress-related symptoms at work [31]. For inclusion, a participant needed to have 1) currently, or within the past two years, been on sick leave due to CMD; 2) received help from the OHS to return to work; 3) experienced that both work- and home-related demands affected their sick leave or RTW and 4) spoken and understood Swedish due to the focus group design. Those who were a victim of bullying in the workplace or had an initial sick leave episode that exceeded three months were excluded.

Data was collected in two parts. During the first phase, all participants included in the original cohort by April 2017 (n = 100) were sent an email with information about the study, which was followed-up by a phone call from the first author (LN). Those who had experienced bullying were provided with referral information to a center that had psychologists with expertise in the subject. Of the initial cohort, 81 employees declined participation, did not meet the inclusion criteria or could not be reached. Due to ethical reasons, explanations for declining to participate were not asked for, but some participants mentioned that they were too tired or did not have time to attend an interview. In total, 14 participants attended the focus group interviews during the first part of the data collection (see Fig 1).

A second recruitment was initiated so that data saturation could be reached, and the number of male participants increased, by placing newspaper ads and distributing information about the study at an OHS center. Those interested contacted the first author (LN) for further details and for the screening of inclusion and exclusion criteria. A short phone interview followed with questions related to their background and health. The second data collection resulted in 14 additional participants (see Fig 1).

Data saturation was deemed complete after deliberations of time constraints, the amount of data and data variation. Six focus groups (n = 3–6) and one pilot interview (n = 2), totaling of 28 participants (13 women and 15 men), were conducted. Two groups had a mixed sex-composition, two groups had only women and two groups had only men. The interview guide was tested during the pilot interview, but as no changes were made and the discussion among the participants was considered rich and interesting, the pilot interview was included in the analysis.

The participants varied in age and educational backgrounds. It was common for the women to work within health care, whereas men mostly worked within engineering, IT and other occupations. Comparing women and men, it was more common for men to be depressed and for women to have anxiety and/or exhaustion, as assessed by the Hospital Anxiety and Depression Scale [32] and the Self-rated Exhaustion Disorder instrument [33]. Moreover, men more often answered that home/family affected work negatively, whereas women more often answered that they were primarily responsible for household work at home. These and other participant characteristics are presented in Table 1.

All participants had received help from the OHS, which was an inclusion criterion to participate in the study. Previous research on OHS interventions in Sweden has described care as usual to consist of work-directed RTW interventions where usually both the employee and the employer are involved together with an OHS consultant [31]. This type of RTW intervention was also mentioned by several of the participants during the interviews. Many also spoke of

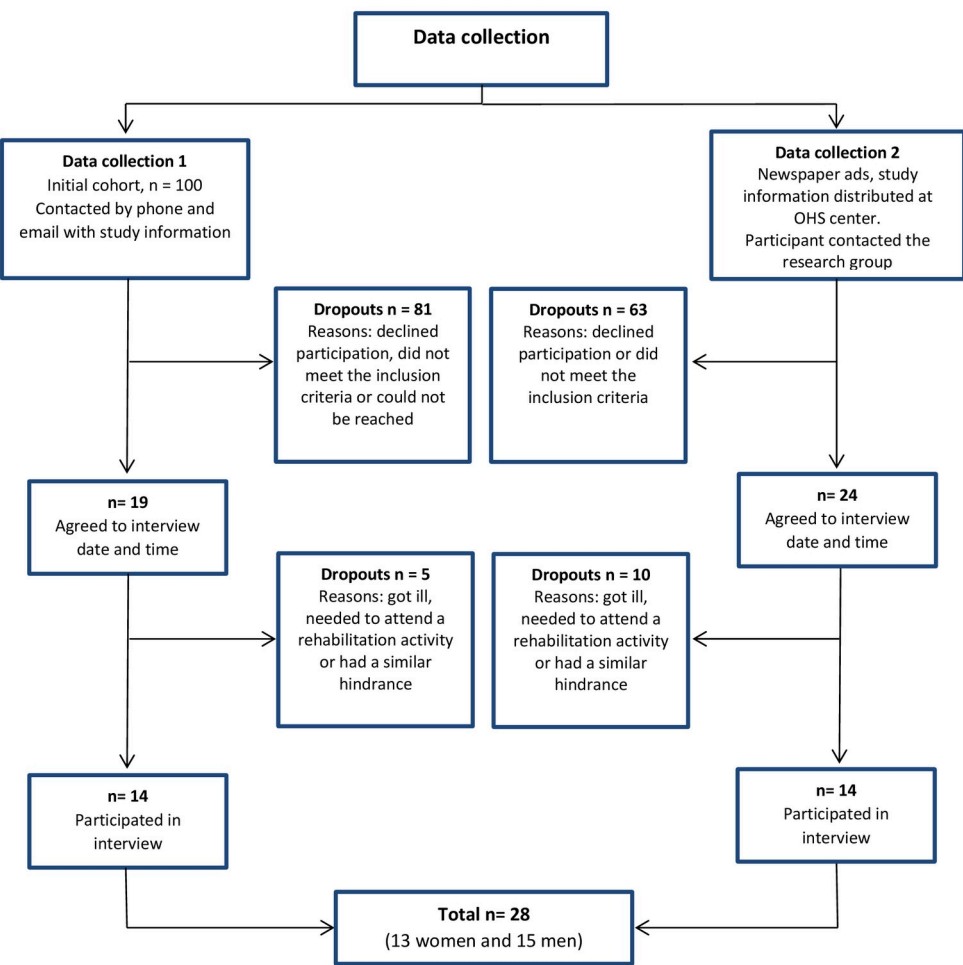

**Fig 1. Flow chart of the data collection.**

gradual return to work; workplace adjustments; cognitive-behavioral therapy; stress management, health or mindfulness courses; written or oral information on sleep, workout or nutrition; physical activity ordered by a doctor; visits to a doctor, psychologist, occupational psychologist, counselor or psychiatrist; and medical treatment for depression or sleeping problems. Most participants also mentioned the sick-leave period as an intervention to return to work. A few spoke of group discussion sessions; visits to a physiotherapist or naprapath; referrals to stress clinics; acupuncture; and medical yoga.

## Ethical considerations

The Regional Ethical Review Board in Stockholm granted ethical permission for the main study (registration number 2015/549–31/1) and the current qualitative part (registration number 2016/972-32 and 2017/2021-32). Amongst other things, it was emphasized that study participation is voluntary and that the participants could withdraw from the study at all stages of the research and without explanation. Before the commencement of the interviews, the participants signed an informed, written consent for study participation and for the interviews to be audio-recorded.

**Table 1. Participant characteristics.**

| | Women (N = 13) | Men (N = 15) |
|---|---|---|
| | N (%) | N (%) |
| **Age** | | |
| 18–29 | 2 (15,4) | 0 (0,0) |
| 30–39 | 1 (7,7) | 1 (6,7) |
| 40–49 | 5 (38,5) | 5 (33,3) |
| 50–59 | 3 (23,1) | 7 (46,7) |
| 60 - | 2 (15,4) | 2 (13,3) |
| **Occupation** | | |
| Health care | 7 (53,8) | 2 (13,3) |
| Engineering/IT | 1 (7,7) | 6 (40,0) |
| Managers/officials | 2 (15,4) | 2 (13,3) |
| Other | 3 (23,1) | 5 (33,3) |
| **Living conditions**[*] | | |
| Alone | 5 (38,5) | 9 (60,0) |
| Lives with: | | |
| Partner | 5 (38,5) | 1 (6,7) |
| Husband/wife | 1 (7,7) | 3 (20,0) |
| Children | 4 (30,8) | 9 (60,0) |
| Other person | 1 (7,7) | 0 (0,0) |
| **Do you have children under the age of 17 living at home** | | |
| No | 9 (69,2) | 7 (46,7) |
| Yes | 4 (30,8) | 8 (53,3) |
| **Highest completed education** | | |
| Primary school or equivalent | 1 (7,7) | 2 (13,3) |
| High school / vocational school | 4 (30,8) | 5 (33,3) |
| University / college education / higher academic degree | 8 (61,5) | 8 (53,3) |
| **How long have you been working at your current workplace?** | | |
| Less than one year | 0 (0,0) | 1 (6,7) |
| 1–5 years | 4 (30,8) | 5 (33,3) |
| 6–10 years | 3 (23,1) | 7 (46,7) |
| More than 10 years | 6 (46,2) | 2 (13,3) |
| **How many days in the last 12 months have you altogether been away from work due to your own illness (sick leave, care, treatment or examination)?** | | |
| 1–7 days | 0 (0,0) | 0 (0,0) |
| 8–24 days | 5 (38,5) | 4 (26,7) |
| 25–99 days | 3 (23,1) | 5 (33,3) |
| 100–365 days | 5 (38,5) | 6 (40,0) |
| **Sick-listed at the time of the interview** | | |
| No | 8 (61,5) | 10 (66,7) |
| Yes | 5 (38,5) | 5 (33,3) |
| **Hospital Anxiety and Depression Scales** | | |
| **Depression** | | |
| No depression | 6 (46,2) | 3 (20,0) |
| Mild depression | 1 (7,7) | 4 (26,7) |
| Depression | 6 (46,2) | 8 (53,3) |
| **Anxiety** | | |
| No anxiety | 2 (15,4) | 5 (33,3) |

(*Continued*)

**Table 1.** (Continued)

|  | Women (N = 13) | Men (N = 15) |
|---|---|---|
|  | N (%) | N (%) |
| Mild anxiety | 6 (46,2) | 4 (26,7) |
| Anxiety | 5 (38,5) | 6 (40,0) |
| **Self-rated Exhaustion Disorder (s-ED)** |  |  |
| No s-ED | 2 (15,4) | 7 (46,7) |
| Mild s-ED | 2 (15,4) | 3 (20,0) |
| S-ED | 9 (69,2) | 5 (33,3) |
| **Who is primarily responsible for the household work at home?** |  |  |
| Myself | 9 (69,2) | 4 (26,7) |
| Someone else | 0 (0) | 0 (0) |
| We divide equally | 4 (30,8) | 11 (73,3) |
| **Does your work affect your home and family life in a negative way?**\*\* |  |  |
| Very rarely or never | 0 (0) | 1 (6,7) |
| Rarely | 0 (0) | 1 (6,7) |
| Occasionally | 5 (38,5) | 7 (46,7) |
| Quite often | 5 (38,5) | 4 (26,7) |
| Very often or always | 2 (15,4) | 2 (13,3) |
| Mean value (SD) | 3,7 (0,8) | 3,3 (1,0) |
| **Does your home / family affect your work in a negative way?**\*\* |  |  |
| Very rarely or never | 3 (23,1) | 3 (20,0) |
| Rarely | 4 (30,8) | 2 (13,3) |
| Occasionally | 5 (38,5) | 5 (33,3) |
| Quite often | 0 (0) | 4 (26,7) |
| Very often or always | 0 (0) | 1 (6,7) |
| Mean value (SD) | 2,2 (0,8) | 2,9 (1,2) |

\*Participants were allowed to choose multiple responses on this question. Some have chosen both "alone" and "living with children", which might reflect that some lived every other week alone, and every other week with their children.

\*\* One of the female respondents answered both "rarely" and "occasionally", and the response was hence excluded.

## Interviews

Informal chat over coffee and snacks took place to create a relaxed atmosphere before the commencement of the interviews. A female moderator (TH in the first interview and LN in the remaining ones) and female co-moderator (LN in the first interview and TH in the remaining ones) participated in all interviews. Their backgrounds and research interests were presented to the participants (LN and TH both hold a PhD, and TH is also an occupational therapist). Both were researchers at the time of the interview and had previous experience as well as training of interviewing and qualitative research. The moderator explained the choice of method, urged the participants to address each other and made sure that everyone could take part in the discussion. The interview guide included areas that corresponded to the aims of the current and the previous study referred to above [28], such as "Experiences of interventions and their effects", "Combined demands at home and at work, and how it affects RTW" and "Have the experiences been affected by being a man or a woman". The interview guide was developed to answer the aims of these studies (see the S1 File for a full account of the included areas and questions). Relevant questions for each area were covered (e.g. "in your experience, did the intervention help you to return to work?") and probes (e.g. "how about the rest of you, did you

experience this similarly or differently?") were used [28]. The interviews took place at an OHS and a research institute in two different cities in Sweden between June 2017 and March 2018. All interviews were audio-recorded, transcribed verbatim and pseudonymised by a professional transcription company. Each interview took between an hour and half to two hours. Field notes were not applied.

### Data analysis

Conventional content analysis was chosen as there is little previous knowledge on the subject [30]. Data analysis begun after completion of all interviews. First, the interview tapes were listened to. Next, the interview transcripts consisting of 192 single-spaced pages were read to gain a sense of the whole and to become immersed in the data [34]. The parts that related to the study aim were highlighted. The interview transcripts were re-read and headings as well as notes were written into the margins until no further aspects could be thought of. These were then transferred to coding sheets and several preliminary categories were thereafter freely derived. This process was done by TH and LN for the first interview and by LN only for the rest of the data. Including two researchers throughout the whole process for the first interview was done so that the emerging categories could be tested, challenged and deepened in order to improve credibility of the analysis. Similarities and differences in the women's and men's experiences were written down in reflective notes during all steps of the analysis. Finally, the derived categories were considered in relation to each other and those considered to belong to the same category were merged and sub-categories were developed until the categories were deemed stable, coherent and distinct from one another. The categories and sub-categories were discussed regularly among all authors to achieve accord of the interpretation. GB is a male professor in occupational health science with basic education in psychology and sociology. IJ is a female professor in methods for corporate health. Alternative interpretations were considered. Feedback of transcripts or findings from the study participants was not applied.

## Findings

The findings illustrate how the male and female participants experienced the help received during their RTW process. One main category and categories at two levels are presented below (see Table 2). Similarities and differences between women and men emerged. For example, it was important for both women and men to be met with respect and recognition, which

**Table 2. Overview of the results.** Main and sub-categories.

| To be met with respect and recognition |
|---|
| *Features in RTW interventions and rehabilitation activities* * |
| Concrete tools to reduce stress |
| Home-related interventions* |
| Individual adaptations at work |
| Organizational interventions* |
| Timing, follow-up and continuity of the help |
| Untangling the various steps to return to work |
| *Responses during RTW interventions and rehabilitation activities* * |
| Having the employee's best interests at heart |
| Professional and skilled treatment by those involved in RTW interventions and rehabilitation activities* |
| An understanding of mental ill health to legitimize the condition* |

*Differences emerged between women and men.

constitutes the main category of the findings. Examples of differences are that women expressed a need for home-related interventions, whereas men did not express such needs as extensively. Furthermore, women and men had different ways of responding to work problems at the organizational level. Whereas women generally made themselves accountable for the situation, men more often demanded that the management take responsibility for it.

## To be met with respect and recognition

The main category is "To be met with respect and recognition". Words such as being "seen" and "understood" occurred frequently in both the women's and the men's descriptions. Key to being satisfied with the help received was that those involved in the RTW process were felt to genuinely consider and validate the participants' experiences. The desire to be respected and recognized was essential to all forms of help that the participants discussed during the focus group interviews, and this was similar for women and men.

Below, the categories "features in RTW interventions and rehabilitation activities" and "responses during RTW interventions and rehabilitation activities" are presented. Further, the first one comprises of six sub-categories that relate to the content and features of interventions and rehabilitation activities. The second contains three sub-categories regarding aspects of other people's responses and own attitudes during the execution of RTW interventions and rehabilitation activities.

**Features in RTW interventions and rehabilitation activities.** Women and men spoke similarly of the importance of concrete tools, adaptations at work during their RTW, correct timing, follow-up and continuity of the interventions, and untangling the various steps to return to work. In addition, women expressed a need for home-related interventions. Both women and men lacked help with organizational problems at work but had different ways of responding to it.

*Concrete tools to reduce stress.* Almost all participants spoke of being helped by concrete tools that, in a systematic manner, slowed down the pace at home and at work, and their use and importance was similar for women and men. The tools were designed to help recognize stress signals, unwind, reduce internal demands, and handle and prevent stress as well as anxieties in recurrent situations in their everyday lives. For instance, mindfulness was used both at work and at home. The participants thought of images that brought them tranquility or aimed to become aware of their immediate surroundings by breathing or listening to the sounds around them: "*Sometimes when I feel overwhelmed, I have a very nice picture of a wonderful mountain top, and I can access the feeling of sitting there and it being completely quite everywhere in the mountains[.] [. . .] My mindfulness became mountain peaks, birds, nature.*" (Woman, pilot interview, sex-mixed group.) They could set an alarm clock that rang after half an hour, signaling that it was time to stop the home-related chore and sit down without doing anything for a while. Other examples on how to unwind were to take a five-to-ten-minute break every hour to let go of everything at work or turned their chairs towards their windows when they needed to rest. The participants described such tools as "common sense", but which were nevertheless difficult to think of on their own. Some participants did not receive such tools and were more restless and unsure of how to handle continued feelings of stress during RTW.

*Home-related interventions.* The interviewed women expressed a need for interventions directed at the home. Although most of the women stated that work-related conditions had been the main reason for their sick leave, they also stated that they needed help with mapping and adjusting demands at home to help ease their total burden: "*When you return to work, you still have your private life. [. . .] It's with you regardless. [. . .] So that would also need to be*

focused on, perhaps . . . 'How does it work for you at home?' Like, maybe tips on how you and your husband could handle it all, or whom I should contact, or how you should talk to your partner, for example." (Woman, focus group 1, all-female group.) The women wanted help with communicating to their partners that they were at reduced capacity and could not take responsibility for household demands in the same way as before, or that they could not do an increased share of household work just because they spent an increased amount of time at home during their sick leave: "I had my husband with me to visit the psychologist, and it was the psychologist who advised me to do that . . . Because we approached the subject of home-related demands and such things. [It was] very valuable." (Woman, focus group 3, sex-mixed group.)

Some of the women had taken their partners to visit psychologists who had helped them understand the women's needs for reduced demands at home; this had been a turning point for many. In a few instances, the interviewed women had not been invited to discuss home-related demands at the OHS as it was not considered a workplace concern. In one example, the woman's main difficulties were related to having children with special needs. She experienced her situation as unsustainable as she had not received help or suggestions for how to manage her home-related demands. In contrast, men did not discuss similar needs for home-related interventions as extensively. They generally found it easier to communicate and create understanding among their partners for their home-related needs during RTW.

*Individual adaptations at work.* It was important for both the male and female participants that their needs for either adjustments or lowered demands at work were met. Examples include reducing the number of assignments, changing work tasks, having the possibility to work from home, getting to keep their own room, participating in supported work or working part-time. There was a need for individual adaptation and for the workplace to recognize the demands that the employee felt were difficult to handle. This was necessary for the employees to be able to find new and sustainable ways of being at work and for combining work, commute, household and children. The employees were instilled with confidence when needs for work adjustments were met, because they knew that they would not have to enter the same stressful work environment upon their return. Those whose needs were not considered at the workplace concluded that they needed to change work, continue their sick leave, retire early or stop doing their jobs well.

*Organizational interventions.* Participants identified several work problems at the organizational level that those involved in their RTW had not been able to help them with. Examples of these were understaffed workplaces, dysfunctional schedules for three-shift work or too many patients or pupils. Consequences included high staff turnover, lack of time to eat lunch, extreme amounts of work, and high rates of sick leave at the workplace. Women and men had different ways of responding to this. Women generally made themselves accountable for the situation and reasoned that they needed to either accept untenable work demands or change jobs: "[I] feel like I won't be able to cope with the demands placed on my workplace, I won't be able to work. I know that, and I have to find someplace else, in fact." (Woman, focus group 3, sex-mixed group.) Men, on the other hand, more often demanded that the management take responsibility over the situation and pitched ideas on how to solve the overarching issues: "At my workplace, there are three or four others who have also ended up in the same situation. [. . .] I kind of try to stress this point with my boss, that it's not just me who has been on sick leave[.] Because they absolutely don't want to take this upon themselves, at my workplace." (Man, interview 4, all-male group.) However, the solutions were seldom considered possible by the management, for which the men felt resignation and contempt: "But no one here is interested in hearing about [suggestions to improve]. [. . .] And this has also made me to feel uncomfortable upon returning. Because now I feel that whatever[.]" (Man, focus group 4, all-male group.) The lack of organizational interventions slowed down both the women's and men's RTW and

made them feel uncertain of whether they could continue working at their jobs in the long run. However, many also expressed that they were too tired to change jobs.

*Timing, follow-up and continuity of help.* Receiving help at the right time and having regular follow-ups was important to reinforce the effectiveness of the interventions during and after RTW for both women and men. Being responsible for themselves, in combination with different "controlling functions" (e.g. physicians, psychologists or Social Insurance Agency contacts) who checked-in on how the participants were doing, increased the participants' motivation and confidence. For example, psychologists gave homework, managers followed-up on plans for work-related adjustments and physicians asked how it was going with physical activity or sleep. In this way, the participants felt that genuine interest was shown them through following-up on previous appointments, instead of treating each meeting as an isolated event. RTW was described by the participants as continuous work with themselves and with demands at work and at home that needed regular follow-ups to help keep them on track. Because the participants' needs changed over time and setbacks were common, the follow-ups made it possible to make necessary adjustments and ensured the applicability of the interventions in their everyday lives:

> "*The treatment you've been through, it's not like the treatment is over. But the treatment is actually a way to introduce yourself to a number of tools, which you then have to continue. . . and consciously use over time. And then no one will be able to say that 'now I'm finished', or 'now I'm healthy'. Instead it's a process that you have given the possibility to use these tools in, to try to reconnect these pieces again.*"

(Man, focus group 5, sex-mixed group.)

The timing of the interventions was also important. For example, almost all participants needed to let go of everything in the initial period of their sick leave in favor of total rest and recuperation, after which they felt the need to actively plan for their RTW. According to the participants, stress management courses and problem-solving discussions were perceived to be more effective after an initial period of rest. Those who received such interventions too early were unable to be receptive, because their cognitive capacities were still hindered:

> *Female participant 1*: "*And then [at the onset of the sick leave period] I was of course so unwell. Because the only thing I could do was to cry. Not because I was depressed, but because I was just so tired, I didn't sleep. [. . .] And in addition they made the mistake of putting me in that group. . . I ended up in a group on stress management. [. . .] And then, I'm not saying it was a total waste, perhaps, but I could have gotten much more out of it today.*"

> *Female participant 2*: "*It was exactly the same for me. I could have sat there and cried all the time, because I was so unwell. It really wasn't the right time to attend [a stress management group], but I did want the information.*"

(Focus group 2, all-female group.)

Similarly, short sick leave spells of two to three weeks were considered counterproductive among both women and men. Participants felt burdened to demonstrate continued ill-health and the need for sick leave during recurring assessments that were conducted three weeks apart. Consequently, some felt compelled to hastily return to work, which, according to the participants, led to repeated and more prolonged spells of sick leave. Those who had received a longer initial sick leave period felt that this was an important component of a successful RTW.

*Untangling the various steps to return to work*. When participants became sick-listed, they needed to manage contacts and tasks related to their sick leave process. Both women and men mentioned that they would have liked a check-list or called upon a contact person who, at the beginning of their sick leave, could help oversee which tasks needed to be done and in what order, whom the participants needed to contact and which things they needed to follow-up. Furthermore, the contact person could give them hands-on help to pursue various tasks such as making phone calls, managing contacts between involved parties or helping them fill out forms The suggestions for the check-list included a list of forms to fill, the rehabilitation options available to the participants and an overview of the rules and regulations at the Social Insurance Agency that the participants felt that were difficult to disentangle:

> Female participant 1: "*I think you need to be able to customize that bit. Just to get. . . a lot of this had suited as a check-list, or just someone who can see the situation in its entirety. [. . .] I mean someone who can package it into a whole.*"

> Female participant 2: "*There are too many loose threads. [. . .] [A check-list or person can prevent] getting into the situation that you rather quit your job, because you don't know where to turn. . . also.*"

> Female participant 3: "*Because it's often that you don't have clarity in all of these different options, because you're in that situation for the first time, and are standing there a bit perplexed.*"

(Focus group 2, all-female group.)

Also, both women and men wanted an outline of the RTW to be able to know what to expect in terms of rest, activity, returning to work and possible set-backs along the way. Participants felt that being able to oversee the situation would reduce feelings of helplessness and confusion in favor of a more focused RTW.

**Responses during RTW interventions and rehabilitation activities.** Participants were fatigued and functioned with a reduced capacity during their RTW. The sub-categories presented below relate either to others' responses or own attitudes that were needed to validate their experiences in order to carry through the RTW interventions and activities. Differences among women and men emerged in the need for knowledge and understanding about mental ill-health. For example, it was easier for women to achieve an understanding of their condition than it was for men, whereas it was easier for men to gain understanding for their need to rest at home than it was for the women. It was important for both women and men to feel that others had their best interests at heart during the RTW process and to experience professional and skilled treatment by those involved in RTW interventions and rehabilitation activities.

*Having the employee's best interests at heart*. Being tired and vulnerable, it was particularly important for the employees to feel that those involved in the RTW process were genuinely concerned about their well-being and this was similar for women and men. Participants were grateful for OHS doctors, managers and psychologists who had time to listen, inquired about their experiences, showed empathy without being patronizing and expressed good will in finding solutions that took the employee's needs into account:

> Male participant 1: "*When you came to the primary care center, ten minutes. So, I get ten minutes, I can't tell you what my situation is in ten minutes?*"

*Male participant 2*: "*Right. But at the primary care center, was it you who. . . they gave me ten, fifteen minutes. During my first meeting with [the OHS], one and a half hour. I was completely shocked. There was someone who took care of me. It. . . it felt just great.*"

(Focus group 4, all-male group.)

However, they expressed insecurities related to their RTW when they were met with frequent changes in appointments, experienced uninterested professionals or professionals who came with "lessons". While participants needed information on mental ill-health, they also wanted it to be communicated in a respectful and validating way, for example by letting them tell how they were doing. To feel that someone had their best interests at heart also meant feeling that the other person was genuinely curious about them. Many felt that the Social Insurance Agency's agents questioned their reasons for sick leave, were curt in their interactions, mistrusted their experiences and were not concerned about their case, which made communication with the agency strained and burdensome. In such instances, the participants felt hindered rather then helped.

Feeling that those involved were (or were not) sincerely considering their interests also determined the extent to which various interventions were experienced as helpful for both women and men. For instance, when the participants discussed workplace-related demands together with the OHS doctor, their manager and at times another representative (e.g. from HR or the union), these meetings were considered useful when the participants' interests were considered, but frustrating and meaningless when profits or other company interests were experienced to overrule them. Similarly, managers who said that they didn't have time for the employees once they were back at work or who did not make recommended work adjustments created uncertainty about the sustainability of RTW among the participants.

Finally, the assurance that others were thinking of the participants' best was ascribed not as much to a choice of words or a specific set of actions as to a general disposition:

"*[I] had like seven doctors in a year, they were all completely uninterested. But they followed these directives that they have, they ask, 'what can we ask of your manager', and 'how can we change your work situation?' You know? But at the same time, I personally feel that they don't actually care how it will work out for me.*"

(Man, focus group 4, all-male group.)

In another example, the demands on an employee had reduced at his workplace, but he was nevertheless anxious because he was not sure if it had been done for his benefit or because his manager no longer trusted his capacity. This was also due to an unresolved conflict that he had with his manager. The male and female participants who felt that others had their best interests at heart and genuinely cared about achieving a successful RTW for the employee, achieved a renewed belief in their capabilities to return to work.

*Professional and skilled treatment by those involved in RTW interventions and rehabilitation activities.* Both women and men felt that there was a lack of understanding of mental ill-health and its consequences in encounters with professionals involved in the RTW. Several of the participants believed that this was due to a lack of knowledge on what mental ill-health is and how vulnerable they were in their condition. Men had wanted more help from professionals (e.g. physicians) with special competence in stress-related symptoms to receive the best care possible, whereas this was seldom mentioned by the women. Both women and men felt that if agents at the Social Insurance Agency would have had more knowledge of mental ill-health, they would have been more respectful in their communication and accurate in their decisions.

The effects of the lack of knowledge were also experienced at work. In one instance, co-workers ridiculed and questioned the employee's need to work shorter days as they did not understand his level of exhaustion. Participants suggested that managers and colleagues be trained and provided with information about mental ill-health and ways of socially supporting employees upon RTW.

Furthermore, almost all participants told their opinion was asked on the length of the sick leave, but they were incapable of making correct assessments, and hence needed help in determining the length of their sick leave period. As they had high demands on themselves and had not yet understood what had happened to them, participants often thought that they needed only a couple of days or weeks until they could be back at work:

> "*My boss is also very good, he's great. [. . .] He compared this with breaking a leg, saying that you don't call someone who has a broken leg to come work the day after [as his HR department had done], it takes time. And he has kept to his plan that he had from the outset, which I did not believe in because I thought that I would be back at work much faster than that, but his plan is correct because it's in sync with how I'm doing.*"

(Man, focus group 6, all-male group.)

Retrospectively, both male and female participants discussed at length how they perceived that they had not been able to evaluate their situation correctly at the onset, and were grateful for the support of physicians or managers who had directed them to longer sick leave periods or had slowed down their RTW when necessary. When the length of the sick leave corresponded to the actual pace of their RTW, participants expressed more confidence for the sustainability of the RTW. However, in cases were physicians were experienced to let the participants' assessments determined the initial length of the sick leave period, participants felt the initial period became too short and didn't give the opportunity for them to let go and relax, leading to repeated, more prolonged sick leave spells and a larger sense of insecurity in relation to the RTW.

*An understanding of mental ill health to legitimize the condition.* In addition to how others responded to the participants during RTW, the participants' own attitude also affected their experiences of rehabilitation activities and RTW interventions. They needed facts and information about mental ill-health to legitimize their condition for themselves and for those around them at work and at home, and in interactions with other involved parties. Knowledge of mental ill-health was often described as a prerequisite for the other components of the RTW, such as communicating and understanding needs and adjusting demands. Knowledge of mental ill-health was experienced as connected to gaining understanding of the condition. When participants had facts and information about mental ill-health, it became easier to understand and accept for themselves and communicate to others that they did not have the same capacities as before and that they needed to make adjustments both at home and at work for a successful RTW. In general, although men needed more help in understanding and accepting that they had mental ill-health, they felt that their partners were relatively quick to understand what they were going through and gave them good emotional as well as practical, household-related (e.g. laundry, cleaning, cooking and to an extent care-taking of children) support:

> "*[I]'ve had fantastic support from my wife. She's had an absolute understanding and helped me during my sick leave[.] In the purely practical, it's about [bringing and leaving kids at school/daycare], food and groceries, planning and the like. I mean. . . especially when I was on*

*sick leave [. . .] and prescribed to rest, exercise and eat. [. . .] To lie on the couch and read a book is quite provocative for a wife who has quite a lot to do. And yet she understood.*"

(Man, pilot interview, sex-mixed group.)

By comparison, women generally accepted and grasped their condition rather quickly, but described greater difficulties in communicating it to their partners in order to receive emotional and practical, household-related support. As compared to the men, it was more common among the interviewed women to receive support from friends who understood their experiences of mental ill-health and validated them. Participants who lacked information about mental ill-health also had more difficulties in bringing together different aspects of their RTW, creating self-doubt about their current condition and capabilities. For example, if they had knowledge of what mental ill-health entailed, it also became easier for the participants to communicate their needs to adjust demands both at work and at home.

## Discussion

Although women are at a higher risk of being absent because of CMD-related sickness than men, few studies have explored whether women and men experience interventions and rehabilitation activities during RTW differently. The present study found both similarities and differences between the experiences of women and men in this regard. During the focus group interviews, both women and men emphasized the importance of being met with respect and recognition irrespective of the form of help received. However, whereas women expressed a need for home-related interventions, for example, men did not express such needs as urgently. Also, whereas women held themselves accountable for work problems at the organizational level, men more often demanded that the management take responsibility for it.

The main category comprised that it was important for the participants to be met with respect and recognition by the professionals that they met during RTW. This finding is consistent with those in several previous qualitative studies, in which interviewees tend to mention words and phrases that are similar to the ones in our study, for example "being seen" and "understood" [25, 35–37]. In addition, our study found that such expectations are expressed to be of equal importance and described in similar terms by women and men. Employees who felt that their experiences were validated also expressed more satisfaction with the help that they received from professionals such as psychologists, case workers and medical doctors, which is also in line with other studies [25, 35, 36, 38]. Previous research has suggested that those with mental ill health may experience encounters with professionals differently from those with other health problems [21, 22, 39]. For example, results from a Swedish report showed that as compared to employees who were on sick leave for other reasons, employees on long-term sick leave for mental ill health were more likely to consider that the perceived quality of their treatment by the officer had an impact on their RTW [39]. Therefore, if employees with CMD would perceive that their experiences are valued in the encounters with professionals, then that could help realize the full potential of RTW interventions. Furthermore, studies among employees with back pain have found that a validating communication technique can help participants reduce their frustration, anger and sadness as compared with participants who have not been treated with this technique [40]. Our results suggest that a similar, active validation of employees' experiences may also be beneficial to both women and men with work-related CMD.

In particular, the participants mentioned the Swedish Social Insurance Agency as central to their experiences of validation when they spoke of their contacts with an external, third party. Barring a few exceptions, the experiences were negative, which is consistent with the findings

of a meta-synthesis of qualitative studies on employees with CMD. This synthesis found that contacts with particularly social insurance agencies were often perceived as difficult. These authorities were described as being more interested in the employee's rapid RTW, instead of showing concern for the employee's mental ill health or needs during the sick leave period [24]. Social insurance agencies need to often engage in a difficult balancing act of adhering to existing regulations while also having to accommodate people with mental ill health in vulnerable situations. Agents at such agencies may therefore need support and training so that they may both implement regulations and meet the needs of people with mental ill health that they are in contact with.

Several participants felt burdened to demonstrate continued ill-health and the need for sick leave during recurring assessments that were made two weeks apart. Some felt this caused them to return to work too early, negatively affecting their RTW in the long run, a finding that is corroborated by other qualitative research [24]. By contrast, those who received a long sick leave spell of six weeks at the start felt that it had been an important factor for a successful RTW. However, when it comes to employees sick-listed because of CMD, improved symptoms do not necessarily lead to improved work ability, or vice versa [6], implying that the extent of the sick leave needs to be considered within RTW interventions. Furthermore, work has a positive impact on health and well-being [41], and sick leave increases the risk of future disability pension, recurrent sick leave and unemployment [42]. Some studies also indicate that interventions initiated in the first 6 weeks of sickness absence are more favorable to RTW than those initiated later [43]. Furthermore, work-focused interventions have been found effectual for partial RTW [6], and a gradual RTW is experienced as an important facilitator to RTW by employees [24]. A more flexible and beneficial solution would thus be shorter spells of sick leave with the possibility of regularly assessing the employee's situation and needs to RTW, including considerations of for example work modification and possibilities for a gradual return, rather than receiving a longer sick leave spell from the beginning. However, while the employees often mentioned such interventions, they also expressed the need for a clear overview of the RTW-process to reduce feelings of helplessness and confusion in favor of a more focused RTW. Thus, the framing and communication of both the assessments and the extent of the sick leave might warrant attention for it to work in favor of the employee. A scoping review found that positive expectations towards the duration of the sick leave or RTW predicted earlier RTW among employees with CMDs than those who did not have such expectations [44]. Considering the employee's expectations and exploring possibilities to affect them might also be beneficial to consider during the framing and communication of the sick leave. Additionally, a previous qualitative study on a multidisciplinary RTW program managed by municipal sickness benefit offices found that assessment consultations could create frustration and uncertainty among employees with CMD [25]. This is because the employee had difficulties in verbalizing their mental condition to RTW professionals. Our results in an OHS context were similar, but they also show that communicating the mental condition was particularly difficult for men, as they had greater difficulties in understanding and accepting their condition as compared with women. This could be a reason why health professionals might not be able to recognize mental ill health in men as effectively as in women [14], and women may be more receptive toward sickness absence in encounters with doctors than with men [13]. Previous literature shows that gender norms affect such experiences and that difficulties in communicating with health professionals seem to result from beliefs and attitudes concerning masculinity [45]. For example, men may not feel comfortable with sharing emotional aspects of their lives as it puts them in what they see as a vulnerable position, thereby clashing with perceptions of more traditional masculinities [45]. Considering the effects of masculinity

norms in encounters with RTW professionals might thus strengthen the focus and effect of RTW interventions and rehabilitation activities.

Most participants expressed the need to continuously work with themselves and the tools that they had received to achieve a long-term RTW. This is consistent with the findings of a previous study that evaluated two different rehabilitation programs for patients on sick leave for burnout. That study concluded that it takes time to implement cognitive tools and establish new behaviors [46]. The previously mentioned meta-synthesis of qualitative studies also found that employees with CMD needed additional time to implement the solutions and coping strategies that they had developed during RTW to maintain their work capacity [24]. Our results suggest that follow-ups and added time were deemed important because this allowed for necessary adjustments and for ensuring the applicability of the interventions in their every-day lives as their needs changed over time. Follow-ups with the possibility of making the required adjustments may thus be needed to achieve the full potential of such tools and for their effects to last over time.

## Differences in experiences of the interviewed women and men

Some important differences between women and men emerged, which confirm previous calls for examining the possible influence of gender on the RTW process [47]. For example, the finding that women tended to hold themselves accountable for work problems at the organizational level is in line with a previous study among female workers with CMD [12]. It is also consistent with the finding of another study among medical trainees with burnout that women used self-blame as a coping strategy more frequently than men, which appeared to cancel other, more adaptive coping skills that the women used. The authors of that study reasoned that this could be one explanation for women's higher rates of burnout as com-pared with men [48]. By contrast, the men we interviewed expressed the need for organiza-tional interventions and often demanded that the management take responsibility. When the management remained inactive despite the men's efforts, however, this negatively affected the men's health and hopes for a full recovery. This also reduced their motivation to work, which has been shown to hinder RTW [49]. These findings imply that to ease their feelings of guilt during RTW interventions and rehabilitation activities, women may need support in mapping how structural work-environment problems affect mental health and in finding solutions to these problems. By contrast, men may need support in finding solutions to structural work-environment problems to counteract feelings of resignation to support long-term RTW.

Gender norms may also help explain the finding that the interviewed women expressed a need for home-related interventions (for example conversation support, increased partner support, strategies for finding moments of rest at home), also when work-related demands were experienced as the main reason for the sick leave. The women felt that home-related demands hindered their RTW, while men described greater emotional and practical support in the home. This corroborates the findings of a study of employees with musculoskeletal problems [26]. They are also consistent with the findings of a systematic review that employees with mental ill health may have greater needs for recovery at home as they need to make extra efforts at work to be productive [17]. Whereas the review did not compare the experiences of women and men, our results imply that men may have more favorable conditions for recovery at home as compared to women. Qualitative [24, 37] and quantitative [6, 15–17, 19] literature reviews on RTW among employees with CMD seldom discuss home-related aspects of the interventions. However, our results suggest that interventions should consider home-related aspects in addition to work-related ones. The results also imply that studies should verify that

their findings are of equal relevance to both women and men, to avoid the risk of gender-biased support during RTW [50].

## Strengths and limitations

Relevance, reflexivity and validity have been proposed as essential criteria to evaluate qualitative research [51]. CMD are an acute health problem with great consequences for the individual, the employer and for society at large, which underscores the relevance of the current study. The focus group interviews in the current study have provided access to the experiences of people on sick leave for CMD, and such interviews could be effective tools for those who work with people in similar circumstances. During the process of analysis, the results were discussed among all authors of this paper, who contested or confirmed the emerging categories, thus improving the reflexivity of the study. Alternate and even opposing interpretations were deliberated. To improve validity, efforts were made to present information about data collection, participant characteristics and analysis in a clear manner. Minority cases were considered and presented in the results. Several quotes were provided to illustrate the findings, with the aim of enhancing transferability of the results to other studies with similar or related research questions. Data saturation was considered complete as the last few interviews did not significantly add to new information with regards to the study aim.

All group interviews were conducted by two women, which is likely to have affected interview interactions and knowledge production [52, 53]. Including a male interviewer might have led to other themes growing stronger or weaker. However, this point was addressed during the analysis of the data when both female researchers (LN, TH, IJ) and a male researcher (GB) participated. For the same reasons, the focus groups were ordered in all-female, all-male and sex-mixed groups. Some differences between these arrangements were indeed noticed. For example, whereas some themes grew stronger in the all-female groups (e.g. the need for home-related interventions), other themes were more pronounced in the all-male groups (e.g. the need for organizational interventions). In the sex-mixed groups, one sex could introduce a topic that the other sex might otherwise not have considered, which affected subsequent analysis. A strength of this study was that not only differences, but also similarities between women's and men's experiences were explored. This allowed to minimize the risk of gender stereotyping [54].

## Conclusion

Similarities as well as differences emerged in how women and men sick-listed because of CMD experienced RTW interventions and rehabilitation activities. It was important for both women and men to be met with respect and recognition, which was essential to all forms of help that the participants discussed during the focus group interviews. Women expressed self-blame and a need for home-related interventions, whereas men expressed a need for organizational interventions to counter feelings of resignation. A key implication of this study is that research on interventions and rehabilitation activities during RTW among employees on sick leave for CMD should consider whether the findings are relevant equally to both women and men. Similarly, RTW professionals may need to consider possible differences among women and men on sick leave for common mental disorders, and to further customize offered interventions and rehabilitation activities. Doing so may help enhance the effectiveness of such interventions.

## Supporting information

**S1 File. Interview guide.**
(PDF)

## Author Contributions

**Conceptualization:** Lotta Nybergh, Gunnar Bergström.

**Formal analysis:** Lotta Nybergh, Gunnar Bergström, Irene Jensen, Therese Hellman.

**Funding acquisition:** Lotta Nybergh, Gunnar Bergström, Irene Jensen.

**Investigation:** Lotta Nybergh, Therese Hellman.

**Methodology:** Lotta Nybergh, Gunnar Bergström, Irene Jensen, Therese Hellman.

**Project administration:** Lotta Nybergh.

**Supervision:** Gunnar Bergström.

**Validation:** Lotta Nybergh, Gunnar Bergström, Irene Jensen, Therese Hellman.

**Visualization:** Lotta Nybergh, Gunnar Bergström, Irene Jensen, Therese Hellman.

**Writing – original draft:** Lotta Nybergh.

**Writing – review & editing:** Lotta Nybergh, Gunnar Bergström, Irene Jensen, Therese Hellman.

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
