## [Decision Letter · Decision Letter 0]

1 Apr 2021

PONE-D-20-33900

Experiences of interventions and rehabilitation activities in connection with return-to-work from a gender perspective. A focus group study among employees on sick leave for common mental disorders.

PLOS ONE

Dear Dr. Nybergh,

Thank you for submitting your manuscript to PLOS ONE. After careful consideration, we feel that it has merit but does not fully meet PLOS ONE’s publication criteria as it currently stands. Therefore, we invite you to submit a revised version of the manuscript that addresses the points raised during the review process.

Please carefully check the manuscript for typos.

We look forward to receiving your revised manuscript.

Kind regards,

Paolo Roma

Academic Editor

PLOS ONE

Journal Requirements:

3. Please include additional information regarding the interview guide or script used in the study and ensure that you have provided sufficient details that others could replicate the analyses. For instance, if you developed a guide as part of this study and it is not under a copyright more restrictive than CC-BY, please include a copy, in both the original language and English, as Supporting Information

5. We noted in your submission details that a portion of your manuscript may have been presented or published elsewhere.

"Table 1. Participant characteristics is also included in a manuscript pending at BMC Public Health. The same table is used because the sample is the same. However, the two manuscripts address different research questions."

Please clarify whether this publication was peer-reviewed and formally published. If this work was previously peer-reviewed and published, in the cover letter please provide the reason that this work does not constitute dual publication and should be included in the current manuscript.

6. Please upload a copy of Figure 1, to which you refer in your text on line 160. If the figure is no longer to be included as part of the submission please remove all reference to it within the text.

Reviewers' comments:

Reviewer's Responses to Questions

**Comments to the Author**

1. Is the manuscript technically sound, and do the data support the conclusions?

Reviewer #1: Yes

Reviewer #2: Yes

2. Has the statistical analysis been performed appropriately and rigorously? 

Reviewer #1: N/A

Reviewer #2: Yes

3. Have the authors made all data underlying the findings in their manuscript fully available?

Reviewer #1: No

Reviewer #2: Yes

4. Is the manuscript presented in an intelligible fashion and written in standard English?

Reviewer #1: Yes

Reviewer #2: Yes

5. Review Comments to the Author

Reviewer #1: Thank you for interesting paper. The paper is easy to follow and the results are presented clearly. The topic is addressed from many perspectives.

My suggestions to improve the manuscript are as follows:

1. All the participants had received treatment at OHS. I suggest that the authors specify what interventions and what kind of treatment the participants had received.

2. In the current study, the participants emphasized that, “longer initial period of sick leave is seen as important component of successful return”.

Earlier studies have summarized the effective elements of return-to-work include interventions with a focus on work, tailored return-to-work plan, and gradual, early return to work. Furthermore, the results on work modification and partial sick leave have been positive.

The authors take into consideration the earlier finding that sick leave may increase the risk of disability pension (row 631). This might specifically be the case with prolonged sick leave, for which alternatives should be found.

I would suggest, that the authors would discuss their findings related to these findings of earlier studies. e.g.

- Was work modification available for participants?

- How about the possibilities for gradual return or partial sick leave?

Reviewer #2: The purpose of this study is to explore how men and women sick listed for common mental disorders experience interventions and rehabilitation activities during return to work as well as gendered differences in these experiences.

Introduction: Well written and concise background on the effects of common mental disorders (CMDs) on work, sickness absence and rehabilitation/RTW with a thorough depiction of how gender differences unfold given women’s domestic responsibilities, greater likeliness to seek help, differences in job characteristics.

Findings: I would consider consolidating a few of the subsections such as “Individual adaptation at work” and “Organizational interventions” under a broader heading as many elements are work-related – although I’ll leave this up to the authors.

Well written discussion with inclusion of prior research in the area that confirms or negates findings. Happy to see a separate section for gender considerations (differences between men and women’s experiences).

Methodological considerations can be renamed Strengths and Limitations.

6. PLOS authors have the option to publish the peer review history of their article (what does this mean?). If published, this will include your full peer review and any attached files.

Reviewer #1: No

Reviewer #2: No

---

## [Author Response · Author response to Decision Letter 0]

29 Apr 2021

Answer to the additional questions provided by the editorial answer after resubmission: 

1) Did the authors present any new data in this submission that were not previously presented in the published article: https://eur01.safelinks.protection.outlook.com/?url=https%3A%2F%2Fdoi.org%2F10.1186%2Fs12889-020-10045-4&data=04%7C01%7Clotta.nybergh%40ki.se%7C87f95d6304404608d59008d90a07668e%7Cbff7eef1cf4b4f32be3da1dda043c05d%7C0%7C0%7C637551849430761648%7CUnknown%7CTWFpbGZsb3d8eyJWIjoiMC4wLjAwMDAiLCJQIjoiV2luMzIiLCJBTiI6Ik1haWwiLCJXVCI6Mn0%3D%7C2000&sdata=4h43wNsHn3z9ZGNvCTEiBvsJIikTlSHk86%2B3ND6%2BH4M%3D&reserved=0?

AUTHOR ANSWER: New data was presented in this manuscript, as the transcribed interviews were analyzed with a different aim compared to the previously published article. Whereas the current manuscript explores how employees on sick leave for common mental disorders experience interventions and rehabilitation activities during return-to-work, the published article explored how the experiences of work- and home-related demands as well as resources influence return-to-work among employees sick-listed for common mental disorders in Sweden. The findings and conclusions of the current study are hence novel.

 2) Did the authors perform any additional experiments or collect any additional data that were not a part of the study from the published article: https://eur01.safelinks.protection.outlook.com/?url=https%3A%2F%2Fdoi.org%2F10.1186%2Fs12889-020-10045-4&data=04%7C01%7Clotta.nybergh%40ki.se%7C87f95d6304404608d59008d90a07668e%7Cbff7eef1cf4b4f32be3da1dda043c05d%7C0%7C0%7C637551849430761648%7CUnknown%7CTWFpbGZsb3d8eyJWIjoiMC4wLjAwMDAiLCJQIjoiV2luMzIiLCJBTiI6Ik1haWwiLCJXVCI6Mn0%3D%7C2000&sdata=4h43wNsHn3z9ZGNvCTEiBvsJIikTlSHk86%2B3ND6%2BH4M%3D&reserved=0?

AUTHOR ANSWER: 

New data was presented in this manuscript, as the transcribed interviews were analyzed with a different aim compared to the previously published article. Whereas the current manuscript explores how employees on sick leave for common mental disorders experience interventions and rehabilitation activities during return-to-work, the published article explored how the experiences of work- and home-related demands as well as resources influence return-to-work among employees sick-listed for common mental disorders in Sweden. The findings and conclusions of the current study are hence novel.

This response has also been outlined in one of the attached files, if it may be easier to read from it: 

PONE-D-20-33900 

Experiences of interventions and rehabilitation activities in connection with return-to-work from a gender perspective. A focus group study among employees on sick leave for common mental disorders.

We would like to thank the academic editor and reviewers for their time as well as insightful comments and suggestions on how to improve the manuscript. We have responded to the points one-by-one below. 

In response to the points raised by the academic editor:

Thank you for your guidance. We have gone through the manuscript in accordance with the above links and made the following formatting changes: 

• The figure and table references have been changed to bold 

• The title has been centered on the first page

• The author affiliations have been indicated in superscript 

• All references have been given in brackets instead of parenthesis

We have ensured that all references are included in the reference list and checked them for accuracy. Some minor edits have been made and one reference that was mentioned twice has now been deleted (and the consequent reference numbers adjusted). We have also gone through the references one-by-one and looked them up on their publishers’ sites but were unable to identify any retracted references. If there are specific reference(s) of concern that you can alert us to, this would be helpful in investigating the matter further. 

3. Please include additional information regarding the interview guide or script used in the study and ensure that you have provided sufficient details that others could replicate the analyses. For instance, if you developed a guide as part of this study and it is not under a copyright more restrictive than CC-BY, please include a copy, in both the original language and English, as Supporting Information

Thank you. We have now reported in the manuscript that the interview guide was developed within this study (see yellow marking below), and we have also included the guide as Supporting Information in both English and the original, Swedish, language: 

“The interview guide included areas that corresponded to the aims of the current and the previous study referred to above (28), such as “Experiences of interventions and their effects”, “Combined demands at home and at work, and how it affects RTW” and “Have the experiences been affected by being a man or a woman”. The interview guide was developed to answer the aims of these studies (see the S1 File for a full account of the included areas and questions). Relevant questions for each area were covered (e.g. “in your experience, did the intervention help you to return to work?”) and (…).”

We have provided the following information in the cover letter on why there are data sharing restrictions: 

“Since the data is based on transcribed interviews, the data cannot be completely anonymized and can therefore not be shared openly; although names, places and the like have been removed from the transcripts, the participants speak of situations and sensitive experiences within specific contexts and with a level of detail that would pose a risk for the participants to be recognized. The participants have also not given their consent for the interview transcripts to be freely shared. Requests by certified researchers for access to the data should therefore be made to the Research and Data Office at the Karolinska Institute, via rdo@ki.se. If permitted by law and ethical approval, which is decided on a case by case basis, the data can be shared.”

5. We noted in your submission details that a portion of your manuscript may have been presented or published elsewhere.

"Table 1. Participant characteristics is also included in a manuscript pending at BMC Public Health. The same table is used because the sample is the same. However, the two manuscripts address different research questions."

Please clarify whether this publication was peer-reviewed and formally published. If this work was previously peer-reviewed and published, in the cover letter please provide the reason that this work does not constitute dual publication and should be included in the current manuscript.

During the current review process, Table 1 has been peer-reviewed and published: Nybergh, L., Bergström, G. & Hellman, T. Do work- and home-related demands and resources differ between women and men during return-to-work? A focus group study among employees with common mental disorders. BMC Public Health 20, 1914 (2020). https://doi.org/10.1186/s12889-020-10045-4 . 

However, the published and the current study have two distinct aims. Whereas the current manuscript explores how employees on sick leave for common mental disorders experience interventions and rehabilitation activities during return-to-work, the published article explored how the experiences of work- and home-related demands as well as resources influence return-to-work among employees sick-listed for common mental disorders in Sweden. We believe that it is meaningful to present the sample information in Table 1 in both manuscripts as it provides context for the findings. However, if you would prefer that we refer to the other article for details on sample characteristics, we are happy to do so.

Furthermore, the English version of the interview guide (see point 3 above) and a sex-segregated version of Figure 1 (see point 6 below) have also been published in the abovementioned manuscript (the interview guide and the sex-segregated version of the figure were added during the review process of that manuscript). In line with the above, if you would prefer that we refer to the published article for these details, we are happy to do so.

6. Please upload a copy of Figure 1, to which you refer in your text on line 160. If the figure is no longer to be included as part of the submission please remove all reference to it within the text.

Thank you for noticing; we have now uploaded the accompanying figure. A sex-segregated version of this figure has also been published in the manuscript mentioned above under point 5, whereas this figure presents the data combined. If you would rather that we republish the same sex-segregated figure from the above-mentioned manuscript, or that we refer to that manuscript directly for these details, we are happy to do so. 

In response to the points raised by the reviewer 1:

1. All the participants had received treatment at OHS. I suggest that the authors specify what interventions and what kind of treatment the participants had received.

Thank you for this relevant comment. The interview guide included a question regarding the type of help and rehabilitation activities that the participants had received to return to work. The following information has now been added to the manuscript: 

“All participants had received help from the OHS, which was an inclusion criterion to participate in the study. Previous research on OHS interventions in Sweden has described care as usual to consist of work-directed RTW interventions where usually both the employee and the employer are involved together with an OHS consultant (34). This type of RTW intervention was also mentioned by several of the participants during the interviews. Many also spoke of gradual return to work; workplace adjustments; cognitive-behavioral therapy; stress management, health or mindfulness courses; written or oral information on sleep, workout or nutrition; physical activity ordered by a doctor; visits to a doctor, psychologist, occupational psychologist, counselor or psychiatrist; and medical treatment for depression or sleeping problems. Most participants also mentioned the sick-leave period as an intervention to return to work. A few spoke of group discussion sessions; visits to a physiotherapist or naprapath; referrals to stress clinics; acupuncture; and medical yoga.”

2. In the current study, the participants emphasized that, “longer initial period of sick leave is seen as important component of successful return”.

Earlier studies have summarized the effective elements of return-to-work include interventions with a focus on work, tailored return-to-work plan, and gradual, early return to work. Furthermore, the results on work modification and partial sick leave have been positive.

The authors take into consideration the earlier finding that sick leave may increase the risk of disability pension (row 631). This might specifically be the case with prolonged sick leave, for which alternatives should be found.

I would suggest, that the authors would discuss their findings related to these findings of earlier studies. e.g.

- Was work modification available for participants?

- How about the possibilities for gradual return or partial sick leave?

Thank you. We have added the following in the manuscript to address this point (the new parts are marked in yellow), in addition to the previous considerations made on this issue:

“Several participants felt burdened to demonstrate continued ill-health and the need for sick leave during recurring assessments that were made two weeks apart. Some felt this caused them to return to work too early, negatively affecting their RTW in the long run, a finding that is corroborated by other qualitative research (24). By contrast, those who received a long sick leave spell of six weeks at the start felt that it had been an important factor for a successful RTW. However, when it comes to employees sick-listed because of CMD, improved symptoms do not necessarily lead to improved work ability, or vice versa (6), implying that the extent of the sick leave needs to be considered within RTW interventions. Furthermore, work has a positive impact on health and well-being (42), and sick leave increases the risk of future disability pension, recurrent sick leave and unemployment (43). Some studies also indicate that interventions initiated in the first 6 weeks of sickness absence are more favorable to RTW than those initiated later (44). Furthermore, work-focused interventions have been found effectual for partial RTW (6), and a gradual RTW is experienced as an important facilitator to RTW by employees (24). A more flexible and beneficial solution would thus be shorter spells of sick leave with the possibility of regularly assessing the employee’s situation and needs to RTW, including considerations of for example work modification and possibilities for a gradual return, rather than receiving a longer sick leave spell from the beginning. However, while the employees often mentioned work modification and gradual return, they also expressed the need for a clear overview of the RTW-process to reduce feelings of helplessness and confusion in favor of a more focused RTW. Thus, the framing and communication of both the assessments and the extent of the sick leave might warrant attention for it to work in favor of the employee. A scoping review found that positive expectations towards the duration of the sick leave or RTW predicted earlier RTW among employees with CMDs than those who did not have such expectations (45). Considering the employee’s expectations and exploring possibilities to affect them might hence be beneficial to consider during the framing and communication of the sick leave. Additionally, a previous qualitative study on a multidisciplinary RTW program managed by municipal sickness benefit offices found that assessment consultations could create frustration and uncertainty among employees with CMD (25). This is because the employee had difficulties in verbalizing their mental condition to RTW professionals. Our results in an OHS context were similar, but they also show that (…).”

Recited or added references within the revised section above: 

• 6. Axén I, Björk Brämberg E, Vaez M, Lundin A, Bergström G, Environmental Health IP. Interventions for common mental disorders in the occupational health service: a systematic review with a narrative synthesis. Int Arch Occup Environ Health. 2020;93:823–838. (Recited)

• 24. Andersen MF, Nielsen KM, Brinkmann S. Meta-synthesis of qualitative research on return to work among employees with common mental disorders. Scand J Work Environ Health. 2012:93-104. (Recited)

• 45. Haitze de Vries AF, Beate Weikert, Alejandra Rodriguez Sanchez & Uta Wegewitz. Determinants of Sickness Absence and Return to Work Among Employees with Common Mental Disorders: A Scoping Review. Journal of Occupational Rehabilitation 2017;28, pages 393–417(2018). (Added)

In response to the points raised by the reviewer 2:

Findings: I would consider consolidating a few of the subsections such as “Individual adaptation at work” and “Organizational interventions” under a broader heading as many elements are work-related – although I’ll leave this up to the authors.

Thank you for raising this interesting point. We have carefully considered the suggestion and revisited the sub-categories to see if they might overlap. However, we were unable to find an overlap for “individual adaptations at work” and “organizational interventions”. Although both include work-related elements, the content of the former describes adaptations specific to the individual to enhance their worker coping strategies to RTW, whereas the content of the latter depicts experiences at the level of the organization, including work environment and perceived sources of stress. Previous research has also found that the individual and organizational levels as well as their interactions merit separate attention in order to facilitate a sustainable RTW for employees with CMDs (e.g. Nielsen K, Yarker J, Munir F, Bültmann U. IGLOO: an integrated framework for sustainable return to work in workers with common mental disorders. Work Stress. 2018. https://doi.org/10.1080/02678373.2018.1438536.) 

We also considered if other categories such as “concrete tools to reduce stress” could be combined with some of the other sub-categories, but similarly concluded that they upheld their stability, coherence and distinctiveness from each other, and we have hence kept the sub-categories intact. 

Methodological considerations can be renamed Strengths and Limitations.

Thank you; we have renamed the section accordingly.

---

## [Decision Letter · Decision Letter 1]

28 May 2021

Experiences of interventions and rehabilitation activities in connection with return-to-work from a gender perspective. A focus group study among employees on sick leave for common mental disorders.

PONE-D-20-33900R1

Dear Dr. Nybergh,

We’re pleased to inform you that your manuscript has been judged scientifically suitable for publication and will be formally accepted for publication once it meets all outstanding technical requirements.

Kind regards,

Paolo Roma

Academic Editor

PLOS ONE

Additional Editor Comments (optional):

Reviewers' comments:

Reviewer's Responses to Questions

**Comments to the Author**

1. If the authors have adequately addressed your comments raised in a previous round of review and you feel that this manuscript is now acceptable for publication, you may indicate that here to bypass the “Comments to the Author” section, enter your conflict of interest statement in the “Confidential to Editor” section, and submit your "Accept" recommendation.

Reviewer #1: All comments have been addressed

Reviewer #2: All comments have been addressed

2. Is the manuscript technically sound, and do the data support the conclusions?

Reviewer #1: Yes

Reviewer #2: Yes

3. Has the statistical analysis been performed appropriately and rigorously? 

Reviewer #1: N/A

Reviewer #2: Yes

4. Have the authors made all data underlying the findings in their manuscript fully available?

Reviewer #1: No

Reviewer #2: Yes

5. Is the manuscript presented in an intelligible fashion and written in standard English?

Reviewer #1: (No Response)

Reviewer #2: Yes

6. Review Comments to the Author

Reviewer #1: Dear Editor / Authors,

Thank you for the revised version. The authors have revised paper accordingly to my comments. I have no further requests.

Reviewer #2: Thank you for addressing the reviewer comments. I feel the discussion is more robust and the inclusion of additional relevant research has strengthened the link between the study findings and prior literature.

7. PLOS authors have the option to publish the peer review history of their article (what does this mean?). If published, this will include your full peer review and any attached files.

Reviewer #1: No

Reviewer #2: No

---

## [Editor Report · Acceptance letter]

16 Jun 2021

PONE-D-20-33900R1 

Experiences of interventions and rehabilitation activities in connection with return-to-work from a gender perspective. A focus group study among employees on sick leave for common mental disorders. 

Dear Dr. Nybergh:

I'm pleased to inform you that your manuscript has been deemed suitable for publication in PLOS ONE. Congratulations! Your manuscript is now with our production department. 

Kind regards, 

on behalf of

Prof. Paolo Roma 

Academic Editor

PLOS ONE